# Outcomes of XEN 63 Device at 18-Month Follow-Up in Glaucoma Patients: A Two-Center Retrospective Study

**DOI:** 10.3390/jcm11133801

**Published:** 2022-06-30

**Authors:** Antonio Maria Fea, Martina Menchini, Alessandro Rossi, Chiara Posarelli, Lorenza Malinverni, Michele Figus

**Affiliations:** 1Struttura Complessa Oculistica, Città Della Salute e Della Scienza di Torino, Dipartimento di Scienze Chirurgiche-Università degli Studi di Torino, 10126 Torino, Italy; alessandro.rossi012309@gmail.com (A.R.); lorenza.malinverni@unito.it (L.M.); 2Department of Surgical, Medical and Molecular Pathology and Critical Care Medicine-University of Pisa, 56126 Pisa, Italy; martina.mmenchini@gmail.com (M.M.); chiara.posarelli@med.unipi.it (C.P.); michele.figus@unipi.it (M.F.)

**Keywords:** XEN63, MIGS, open-angle glaucoma, minimally invasive glaucoma surgery

## Abstract

Background: Glaucoma surgery has significantly evolved over the last years. This paper aims to evaluate the midterm clinical outcomes of the XEN63 device in a real-world scenario. Methods: A retrospective clinical study was conducted on consecutive patients who underwent an XEN63 implant insertion, either alone or in combination with phacoemulsification. The primary endpoint was the mean intraocular pressure (IOP) at the month 18 visit. Results: Twenty-three eyes (23 patients) were included in the analysis. The preoperative IOP was significantly lowered from 27.0 ± 7.8 mmHg to 14.1 ± 3.4 mmHg at month 18 (*p* < 0.0001). At month 18, 14 (77.8%) and 11 (61.1%) eyes had an IOP ≤ 16 mmHg and ≤ 14 mmHg, respectively, without ocular hypotensive medication. The mean number of ocular hypotensive medication taken was significantly reduced from 2.3 ± 0.9 drugs at baseline to 1.0 ± 1.4 drugs at month 18 (*p* = 0.0020). Four (17.4%) eyes had hypotony at postoperative day 1, which was successfully resolved without sequelae. Four (17.4%) eyes underwent a needling procedure and four (17.4%) eyes underwent additional surgeries. Conclusions: The XEN63, either alone or in combination with cataract surgery, significantly lowered the IOP and reduced the need for ocular hypotensive drugs over a period of 18 months.

## 1. Introduction

Glaucoma surgery has experienced important advances over the last several years [1]. Among them, micro- or minimally invasive glaucoma surgery (MIGS) devices have emerged with the aim of providing a safer and less traumatic surgical procedure, while maintaining a good intraocular pressure (IOP)-lowering effect [1,2].

Under the umbrella of the term MIGS, different devices have been included that share a series of common characteristics, such as an adequate IOP reduction with reduced tissue destruction, a good safety profile, shorter surgical time, and rapid postoperative recovery [2,3,4]. However, the criteria for defining a MIGS device have been somewhat controversial [2,5] and the generally accepted definition of MIGS has been modified over the years [6].

Although there may be certain objections when considering the XEN device as a MIGS [2], for the purposes of this document, the term MIGS applies to it.

The evidence evaluating the effectiveness of the XEN63 implant is very limited [7,8,9,10,11], and most studies were performed with an earlier version of the device that was never marketed, possibly due to its high rate of perioperative complications [7,8,9,10].

The new XEN63 device uses the same needle injector as the XEN45 for preventing early side flow and hypotony [11].

We recently published a study, which mainly focused on the safety profile of this device in the early postoperative period (3 months) [11].

The main purpose of the current study is to evaluate the midterm effectiveness and safety of the XEN63 device in patients with glaucoma in a real clinical setting.

## 2. Materials and Methods

### 2.1. Design

A retrospective, open-label, and bicenter clinical study was conducted on consecutive patients who underwent an XEN63 implant insertion, either alone or in combination with phacoemulsification.

The study protocol was approved by the Ethics Committee of the University of Torino, which waived the need for written informed consent. The study was conducted in compliance with the Declaration of Helsinki principles and with Council for Harmonization Good Clinical Practice guidelines.

### 2.2. Patients

The study was conducted on consecutive glaucoma patients who underwent an XEN63 implant insertion, either alone or in combination with cataract surgery, between February and June 2020.

Patients included in the study were aged ≥ 40 years, had a clinical diagnosis of glaucoma, and an unmet target IOP despite medical therapy. Patients with narrow-angle glaucoma were included if, in the surgeon’s opinion, there was enough space to implant the device safely. Patients with closed-angle glaucoma; severe conjunctival problems; phacodonesis; progressive retinal or optic nerve disease of any cause; or history of major ocular surgery (except phacoemulsification) within the previous 6 months were excluded from the study.

### 2.3. Device

The MIGS device used in this study was the XEN^®^63 (Allergan Plc., Dublin, Ireland). It is composed of porcine gelatin crosslinked with glutaraldehyde; it is 6.0 mm long with an outer diameter of 180 μm, and an inner diameter of 63 μm.

### 2.4. Surgical Technique

All the surgical procedures were performed under local anesthesia by the same two surgeons (AMF and MF) who were experienced in this type of surgery.

An intra-Tenon mitomycin-C (MMC) injection (0.1 mL of MMC 0.02–0.03%) was administered into the superonasal quadrant prior to surgery.

The device was placed in the superior nasal quadrant using a standard ab interno technique [11,12,13]. 

The surgical technique was meticulously detailed in a previous paper [11].

In summary, the preloaded injector needle was inserted at the inferotemporal quadrant through a 1.8 mm corneal paracentesis. The final position of the implant was checked with gonioscopy before removing the viscoelastic material.

Sideways movements of the implant were performed, until we were able to verify that the implant could move freely under the conjunctiva, to confirm the lack of adhesion.

In those eyes which underwent a combined surgery (XEN + cataract), XEN63 was implanted in all cases after phacoemulsification.

Perioperative care included antibiotic therapy 4 times a day during 1 week and anti-inflammatory therapy with steroids 6 times daily, which was slowly tapered over three months.

IOP-lowering medications, both topical and systemic, were suspended on the day of surgery.

Needling procedures were performed in the operating room in those eyes with bleb fibrosis, flat bleb, and/or elevated IOP. Needling was the first-choice treatment in these patients. The study protocol allowed up to two needling procedures per eye, after which we proceeded to either use medication and perform a revision or further surgery, as appropriate.

### 2.5. Outcomes

The primary endpoint was the mean IOP at the last follow-up visit.

Secondary endpoints included the mean IOP lowering from preoperative values; reduction in number of ocular hypotensive medications from baseline; proportion of eyes achieving a final IOP ≤ 12 mmHg; ≤ 14 mmHg; ≤ 16 mmHg; ≤ 18 mmHg; or ≤ 21 mmHg with and without medications; and incidence of adverse events.

Success was defined as a month 18 IOP ≤ 16 mmHg without (complete success) or with (qualified success) hypotensive medication and an IOP lowering from preoperative values of at least 30%.

### 2.6. Statistical Analysis

A statistical analysis was performed using Prism 9 version 9.0 (GraphPad Software; San Diego, CA, USA) and MedCalc^®^ Statistical Software version 20.104 (MedCalc Software Ltd., Ostend, Belgium; https://www.medcalc.org (accessed on 28 March 2022)).

Because a sample size calculation was not performed, we conducted a post hoc power analysis for an alpha level of 0.05; the study sample size and the effect size were observed in the study [14].

Data were expressed as number (percentage), mean ± standard deviation (SD), or mean (95% confidence interval, CI) as appropriate.

The Shapiro–Wilks test was used for assessing the normal distribution of quantitative variables.

Changes in IOP and number of ocular hypotensive medications were performed by means of repeated measures ANOVA and the Greenhouse–Geisser correction test.

The last-observation-carried-forward method was used to impute missing data.

Complete success survival rate was plotted using a Kaplan–Meier analysis.

Data for patients who underwent an additional glaucoma surgery were censored at the time of surgery.

*p* value of less than 0.05 was considered significant.

## 3. Results

A total of 23 eyes (23 patients) was included in the analysis, 20 (87.0%) eyes underwent XEN alone, and 3 (13.0) eyes underwent combined surgery (XEN + phacoemulsification). In the overall study sample, the mean age was 67.8 ± 15.3 years, 15 (65.2%) were women, and 14 (60.9%) had a clinical diagnosis of primary OAG. Table 1 summarizes the main demographic and clinical characteristics of the study population.

In the overall study sample, preoperative IOP was significantly lowered from 27.0 ± 7.8 mmHg to 14.1 ± 3.4 mmHg at month 18 (mean difference −12.9 mmHg; 95 CI: −16.9 to −8.9 mmHg; *p* < 0.0001) (Figure 1).

Similarly, preoperative IOP was significantly reduced from 26.5 ± 8.2 mmHg and 30.3 ± 3.3 mmHg to 14.2 ± 3.7 mmHg and 13.7 ± 1.5 mmHg in the XEN alone and XEN + phacoemulsification groups, respectively (*p* < 0.0001 and *p* = 0.0012, respectively).

Data were plotted from the baseline IOP on the *X*-axis and the month 18 visit IOP on the *Y*-axis to determine an overall visual assessment (Figure 2).

At month 18, 14 (60.9%) and 11 (47.8%) patients had an IOP ≤ 16 mmHg and ≤ 14 mmHg, respectively, without ocular hypotensive medication (Table 2).

According to the Kaplan–Meier survival analysis, complete success occurred in 60.9% of eyes (Figure 3).

The mean number of ocular hypotensive medications used was significantly reduced from 2.3 ± 0.9, 2.3 ± 1.0, and 2.3 ± 0.6 drugs at the preoperative visit to 1.0 ± 1.4, 0.8 ± 0.9, and 0.0 ± 0.0 drugs at month 18 in the overall, XEN solo, and XEN + phaco samples, respectively (*p* = 0.0020; *p* = 0.0002; and *p* = 0.0027, respectively).

### Safety

Regarding safety, four (17.4%) eyes had hypotony (defined as an IOP ≤ 6 mm Hg) at postoperative day 1, which was successfully resolved without sequelae. One (4.3%) eye had anterior chamber bleeding during the surgery, one (4.3%) eye had a 1.5 mm hyphema at postoperative day 1, and four (17.4%) had choroidal detachment (three at postoperative day 7 and one at postoperative day 15), which was successfully resolved with medical treatment, at month 1 visit.

Four (17.4%) eyes underwent a needling procedure (mean time for needling 42.9 ± 11.2 days), although only one eye did so due to an elevated IOP.

Four eyes underwent additional surgeries, two (8.7%) eyes underwent trabeculectomy, one (4.3%) eye underwent a device replacement (a new XEN45 device was implanted), and one eye (4.3%) underwent a high-intensity focused ultrasound cyclodestruction (HIFU) procedure (Table 3). However, according to the last follow-up visit, another patient needed either a needling or additional surgery.

The BCVA did not change over the course of the study (mean change: 0.1 ± 0.2).

At month 18, one (5.6%) eye showed a ≥ 2-line worsening in BCVA as compared to baseline, while five (27.8%) eyes showed a ≥ 2-line improvement as compared to baseline (Table 4).

## 4. Discussion

Although glaucoma treatment must follow a patient-tailored approach, in most cases, medical treatment represents the first therapeutic step [15]. However, not all patients achieve and/or maintain adequate glaucoma control with medical therapy [16,17]. In such cases, laser therapy and/or surgery represents the treatment of choice [15,18].

Current scientific evidence with the XEN63 device is very limited [7,8,9,10,11] and, in most cases, provides information about a previous device that was never commercially available [7,8,9,10].

The main difference between the current XEN63 device and the former one is the surgical technique. The new XEN63 device uses a 27G injector, which has the advantage of minimizing the side flow, and, in addition, it is inserted in the anterior chamber through a 1.8 mm corneal paracentesis (the former one used a 25G injector that required a 2.2 mm paracentesis). On the other hand, the studies evaluating the former XEN63 device were performed without MMC [7,8,9,10].

The results of the studies evaluating the former XEN63 device have pointed in the same direction, indicating its good efficacy and safety profile [7,8,9,10]. However, due to the differences between both devices and the surgical technique, it is extremely difficult to compare our results with those of the previous XEN63 studies.

According to the results of our study, the XEN63 device significantly lowered the IOP and reduced the number of ocular hypotensive medications during a follow-up period of 18 months.

In addition, the rate of adverse events was relatively low; the majority of them were mild in severity and were resolved without sequelae (with or without medication).

As far as we know, this is the first paper evaluating the midterm effectiveness and safety of the new XEN63 device.

In a previous paper published by our group, the short-term (3 months) efficacy and safety profiles of this device were analyzed [11]. The results of that study revealed the good hypotensive profile of the device (mean IOP lowering of 40.8 ± 23.5% from preoperative values) and its low incidence of adverse events [11]. Although when compared with the results of our previous study, the IOP increased slightly between month 3 and month 6; from month 6 it remained stable.

Up to now, only XEN45 has been available on the market. The available scientific evidence has shown the mid- and long-term good efficacy and safety profile of this device. When comparing the results of the current study to those of the XEN45 device, it can be observed that the IOP lowering achieved with XEN63 was consistently greater (see Table 5). Moreover, except for the Hengerer et al. study [19], the preoperative IOP of our study was higher than that reported by the other studies [12,20,21,22,23,24,25,26,27,28,29,30]. 

In the current study, four (21.7%) eyes required additional surgery (two eyes underwent trabeculectomy, one eye underwent an XEN45 device implantation, and another one a HIFU procedure). These eyes were those with the highest preoperative IOP (median: 37.0 mmHg; IqR: 26.0 to 41.0 mmHg), which could indicate the need for more aggressive procedures for treating these patients.

Regarding safety, the most commonly reported adverse event was choroidal detachment (four eyes), which was successfully resolved without treatment at the month 1 visit.

Hypotony, defined as an IOP < 6 mmHg, was observed in four (17.4%) eyes at postoperative day 1. In all cases, it was resolved without consequences. The incidence of hypotony reported in studies performed with the former device was similar to ours [8,10], although it should be highlighted that they did not use MMC. In our study, the effect of MMC (favoring hypotony) might have been compensated by the use of a smaller needle.

In our study, postoperative hypotony was not related to ocular complications or visual acuity loss. In the eyes which underwent the XEN63 implant alone, four eyes experienced visual acuity worsening ≥ 1 line, whilenine9 eyes experienced visual acuity improvement ≥ 1 line.

Glaucomatous damage is defined by retinal ganglion cell loss and their axons, although the exact mechanism of retinal ganglion cell death has not been fully elucidated. Once apoptosis has occurred, it is difficult to conceive how visual function might improve. However, before retinal ganglion cell death occurs, they might undergo a period of reversible dysfunction [31]. This hypothesis has been supported by Ventura et al. [32], who found a disproportionate reduction in the pattern electroretinogram amplitude compared to retinal nerve fiber layer thickness. Additionally, a visual acuity improvement following trabeculectomy has been previously reported [33,34,35].

With regard to visual acuity worsening, Lenzhofer et al. [36] reported a BCVA loss of ≥ 2 lines in eight (15%) and two (4%) eyes at month 12 and month 24 after XEN45 implantation. Similarly, Schargus et al. [37] observed a BCVA loss of ≥0.2 lines in nine (5.9%) eyes treated with the XEN45 microstent.

Regarding needling, in the current study, four (17.4%) eyes underwent postoperative needling, which was a lower rate compared to the one reported in previous XEN45 papers [12,20,23,24,25,27,28,30].

When interpreting the results of this study, it is necessary to take into account several limitations. The main one was its retrospective design. Retrospective studies are associated with bias and confounding factors, although the inclusion/exclusion criteria used in our study might limit their impact. The second limitation was the small sample size. Although the sample size was not calculated prior to the study, the power for detecting the observed differences in IOP and the number of ocular hypotensive medications, between preoperative and month 18 values, was 99% and 96%, respectively. Additionally, only three eyes underwent combined surgery (XEN63 + phacoemulsification), which limited the interpretation of the results. Finally, the lack of a control group and its open label design were limitations to consider.

## 5. Conclusions

According to the results of the current study, XEN63 significantly lowered the IOP and reduced the need for ocular hypotensive drugs over a period of 18 months. Additionally, the low rate of complications (the majority of them mild in severity) makes this procedure a safe option.

Different issues, including the impact of the MMC dose on the outcomes, potential factors associated with the procedure success, or whether it may be considered a cost-effective procedure, deserve larger and more controlled trials.

## Figures and Tables

**Figure 1 jcm-11-03801-f001:**
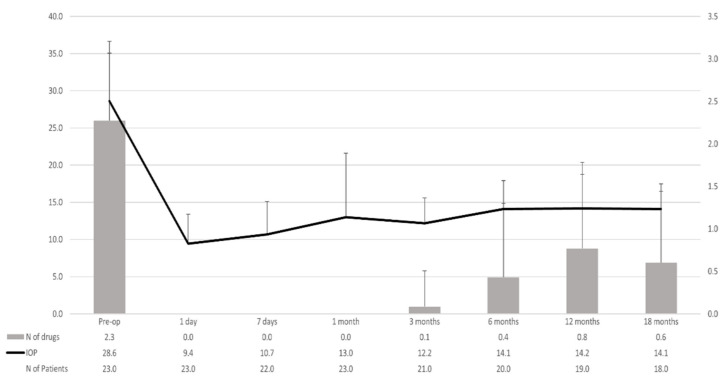
Mean intraocular pressure (IOP) and number of ocular hypotensive medications throughout study follow-up in the overall study population. Vertical bars represent standard deviation.

**Figure 2 jcm-11-03801-f002:**
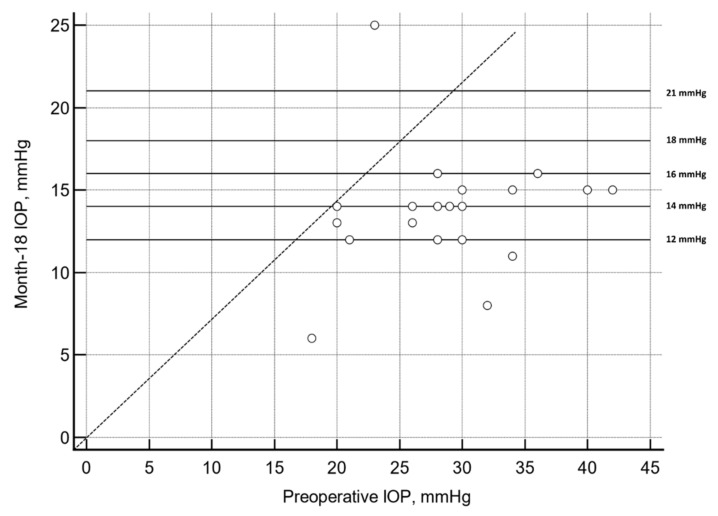
Scatter plot of the preoperative and month 18 intraocular pressure (IOP) (mean difference −12.9 mmHg; 95 CI: −16.9 to −8.9 mmHg; *p* < 0.0001, repeated measures ANOVA and the Greenhouse–Geisser correction test).

**Figure 3 jcm-11-03801-f003:**
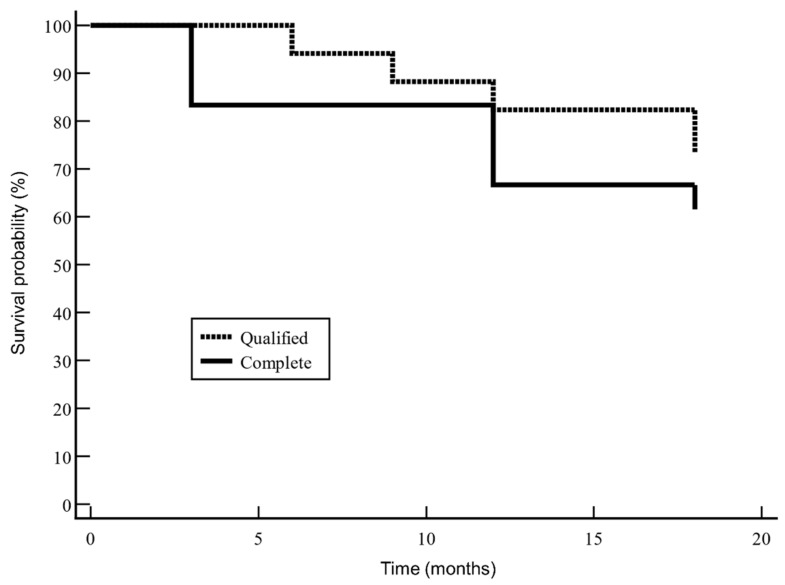
Kaplan–Meier survival curve for complete success. Complete success occurred in 60.9% of eyes, while qualified success occurred in 73.9%.

**Table 1 jcm-11-03801-t001:** Main baseline clinical and demographic characteristics of the study sample.

Variable	Overall (*n* = 23)	XEN 63 (*n* = 20)	Phaco + XEN63 (*n* = 3)
Age, years			
Mean ± SD	67.8 ± 15.3	67.3 ± 15.9	71.3 ± 12.4
Sex, *n* (%)			
Women	15 (65.2)	14 (70.0)	1 (33.3)
Men	8 (34.8)	6 (30.0)	2 (66.7)
Type of glaucoma			
POAG	14 (60.9)	12 (60.0)	2 (66.7)
Uveitic	4 (17.4)	3 (15.0)	1 (33.3)
PXG	1 (4.3)	1 (5.0)	0 (0.0)
PACG	1 (4.3)	1 (5.0)	0 (0.0)
Traumatic	1 (4.3)	1 (5.0)	0 (0.0)
Missing information	2 (8.7)	2 (10.0)	0 (0.0)
Previous laser, *n* (%)			
No	21 (91.3)	18 (90.0)	3 (100.0)
SLT	1 (4.3)	1 (5.0)	0 (0.0)
Nd:YAG iridotomy	1 (4.3)	1 (5.0)	0 (0.0)
Previous surgery *, *n* (%)			
None	2 (8.7)	0 (0.0)	3 (100.0)
Cataract	14 (60.9)	14 (70.0)	0 (0.0)
Refractive (laser)	3 (13.0)	3 (15.0)	0 (0.0)
Trabecular MIG	2 (8.7)	2 (10.0)	0 (0.0)
Subconjunctival MIG	2 (8.7)	2 (10.0)	0 (0.0)
BCVA, ETDRS			
Mean ± SD	0.49 ± 0.26	0.54 ± 0.24	0.23 ± 0.15
ECC			
Mean ± SD	2217. 9 ± 343.1	2223.4 ± 297.0	2161.0 ± 563.8
MD, dB			
Mean ± SD	−17.03 ± 9.96	−16.71 ± 10.51	−18.23 ± 9.44
PSD, dB			
Mean ± SD	7.00 ± 2.28	7.10 ± 3.08	6.66 ± 3.2
NTOHM			
Mean ± SD	2.27 ± 0.94	2.26 ± 0.99	2.33 ± 0.58
IOP, mm Hg			
Mean ± SD	28.7 ± 6.44	28.4 ± 6.81	30.3 ± 3.2

* Patients may have undergone more than one procedure. Abbreviations: Phaco: phacoemulsification; SD: standard deviation; POAG: primary open-angle glaucoma; PXG: pseudoexfoliative glaucoma; PACG: primary angle-closure glaucoma; SLT: selective laser trabeculoplasty; YAG: neodymium-doped yttrium aluminum garnet; MIG: minimally invasive glaucoma device; BCVA: best-corrected visual acuity; ETDRS: early treatment diabetic retinopathy study; ECC: endothelial cell count; MD: mean defect; PSD: pattern standard deviation; NTOHM: number of topical ocular hypotensive medications; IOP: intraocular pressure.

**Table 2 jcm-11-03801-t002:** Overview of the proportion of eyes achieving specific intraocular pressure levels and preoperative IOP lowering ≥30% at month 18, with (qualified) and without (complete) hypotensive medication.

	Qualified	Complete
≤12 mmHg	21.7	21.7
≤14 mmHg	52.2	47.8
≤16 mmHg	73.9	60.9
≤18 mmHg	73.9	60.9
≤21 mmHg	73.9	60.9

**Table 3 jcm-11-03801-t003:** Overview of the postoperative interventions over the course of the study follow-up.

Postoperative Intervention, *n* (%)	
Needling ^a^	4 (17.4)
MMC ^b^	1 (25.0)
5-FU ^b^	3 (75.0)
Digital ocular massage	5 (21.7)
XEN removal ^c^	1 (4.3)
YAG laser ^d^	1 (4.3)
Trabeculectomy ^e^	2 (8.7)
HIFU	1 (4.3)

^a^ Mean time for needling 42.9 ± 11.2 days. ^b^ The percentages were calculated on the basis of eyes which underwent needling. ^c^ An XEN45 device was implanted ab interno. The IOP before removal was 42 mmHg on 3 ocular hypotensive medications. ^d^ To open the XEN63 device by retracting the iris. ^e^ Performed at month 13 and month 14 after XEN implant procedure, respectively. The IOP before trabeculectomy was 36 mmHg on 3 ocular hypotensive medications. HIFU: high-intensity focused ultrasound cyclodestruction. The IOP before HIFU was 40 mmHg on 3 ocular hypotensive medications.

**Table 4 jcm-11-03801-t004:** Proportion of eyes which experienced changes in best-corrected visual acuity throughout the study. Changes in visual acuity were only assessed in eyes which underwent XEN implant alone.

	Month 18
Worse ≥ 2 lines, *n* (%)	1 (5.6)
Worse ≥ 1 line, *n* (%)	3 (16.7)
Unchanged, *n* (%)	2 (11.1)
Improvement ≥ 1 line, *n* (%)	4 (22.2)
Improvement ≥ 2 lines, *n* (%)	5 (27.8)

**Table 5 jcm-11-03801-t005:** A comparison of the clinical outcomes between the current study and the available evidence.

Study	Preop IOP, mm Hg	M12 IOP, mm Hg	M12 IOP Lowering	M 24 IOP, mm Hg	M 24 IOP Lowering, mm Hg	Mean Preoperative NOHM	Mean NOHM, Last Visit	Needling Rates at Last Follow-Up Visit, *n* (%)
Reitsamer et al. [21]	21.4 (3.6) *	14.9 ± 4.5 *	−6.5 ± 5.3 *	15.2 ±4.2 *	−6.2 ± 4.9 *	2.7 ± 0.9 *	1.1 ± 1.2 *	83 (41.1)
Marcos-Parra et al. [22]	19.1 (5.4) *	N.A.	−6.7 (−12.9 to −0.5) **	N.A.	N.A.	2.5 (0.8)	0.2 ± 0.6 *	13 (20.0)
Fea et al. [12]	23.9 (7.6) *	15.5 ± 3.9 *	−7.4 ± 7.9	N.A.	N.A.	3.0 (1.0)	0.5 ± 1.0 *	79 (46.2)
Laborda-Guirao et al. [23]	21.0 (5.2) *	14.7 (13.9 to 15.4) **	−6.3 (−8.8 to −4.4) **	N.A.	N.A.	2.8 (2.7 to 3.0) **	1.1 (0.8 to 1.3) **	7 (8.8)
Gabbay et al. [24]	22.1 ± 6.5 *	15.4 ± 5.9 *	−6.7 ± 6.2 *	14.5 ± 3.3 *	−7.6 ± 5.2	2.77 ± 1.1 *	0.5 ± 1.0 *	57 (37.7)
Mansouri et al. [25]	20.0 ± 7.5 *	N.A.	N.A.	14.1 ± 3.7 *	−6.4 ± 5.9 *	2.0 ± 1.3 *	0.6 ± 0.9 *	58 (45)
Grover et al. [26]	25.1 (3.7) *	15.9 ± 5.2 *	−9.1 (−10.7 to 7.5) **	N.A.	N.A.	3.5 (1.0)	1.7	21 (32.3)
Ibáñez-Muñoz et al. [27] ^1^	22.3 (21.0–23.5) **	15.3 (14.3–16.3) **	−7.3 (−9.7 to −5.0) **	N.A.	N.A.	3.0 ± 1.0	1.2 ± 1.2	19 (26.0)
Theilig et al. [19]	24.5 ± 6.7 *	16.6 ± 4.8 *	N.A.	N.A	N.A	3.0 ± 1.1 *	1.4 ± 1.5 *	42 (42.0)
Hengerer et al. [28]	32.2 (9.1) *	14.2 ± 4.0	32.2	N.A.	N.A.	3.1 ± 1.0 *	0.3 ± 0.7 *	67 (27.7)
Wanichwecharungruang and Ratprasatporn [29]	21.6 ± 4.0	15	30.6	14.6 ± 3.5 *	32.4	2.1 ± 1.4	0.5 ± 0.7 *	10 (17.5)
Subaşı et al. [30]	20.4 ± 4.8 *	15.0 ± 1.9 *	−6.2 ± 0.9 *	14.8 ± 1.9 *	−6.4 ± 1.2 *	3.1 ± 1.0 *	0.9 ± 1.1 *	13 (43.3)
Rauchegger et al. [31]	23.4 ± 7.9 *	14.6 ± 3.6 *	31(20–42)	14.8 ± 4.4	29(30–41)	2.7 ± 1.1 *	1.0 ± 1.2 *	37 (62)
Current study	27.0 ± 7.8 *	14.2 ± 4.6 *	−12.8 (−16.7 to −8.7) **	14.1 ± 3.4 *^, 2^	−12.9 (−16.9 to −8.9) **^, 2^	2.3 ± 0.9 *	1.0 ± 1.4 *	4 (17.4%)

^1^ Mean IOP lowering was −7.3 (−9.7 to −5.0) mm Hg in the primary open-angle glaucoma patients and −6.6 (−8.4 to −4.8) mm Hg in the secondary open-angle glaucoma eyes. ^2^ Month 18 IOP. † Data about standard deviation were not provided. * Mean (standard deviation). ** Mean (95% confidence interval).

## Data Availability

The data presented in this study are available on reasonable request from the corresponding author.

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
