# Peer review of "Outcomes of XEN 63 Device at 18-Month Follow-Up in Glaucoma Patients: A Two-Center Retrospective Study"

_jcm, 2022, doi:10.3390/jcm11133801_

Round 1
Reviewer 1 Report
The current report presents the retrospective IOP and medication reduction data in 23 patients 18 months after undergoing an ab-interno placement of the Xen63 implant with subconjunctival MMC by two surgeons. The reports demonstrates a sustained IOP lowering and reduction in IOP lowering medications that was sustained at 18 months post-op with a rate of secondary glaucoma interventions similar to other Xen device studies.
The paper has several punctuation and grammatical errors related to translation that require profession English editor review and editing.
Materials and Methods:
Patients: Wording of the inclusion and exclusion criteria contains a long run-on sentence. This makes the criteria used by the authors for patient selection confusing and unclear.
Surgical Technique
the authors do not present the dose of MMC administered, only the concentration of MMC solution injected into the subconjunctival space. MMC dose would be important in comparison of other published data on the Xen device.
Results
Figure 3 Kaplan-Meier analysis should have a title and description added similar to the other figures and table
Safety:
Total number of patients requiring a secondary intervention should be reported. According to table 4, secondary interventions may have been required in 14 of the 23 patients. This would represent 61% of patients requiring a secondary interventions after initial Xen63 placement. The authors should further discuss the high rate of needing additional interventions or if a few patients required several of the secondary interventions reported.
Discussion
First paragraph briefly discusses the management of IOP lowering with drops followed but surgery but neglects to mention SLT as an additional option in management.
On line 246-248, the authors state "This may be an important point, as the larger caliber of the Xen 63 device could by clinically meaningful for patients with high preoperative IOP." This statement should be removed as the present study was not designed to compare results with the Xen45. If there is a difference in the results between the Xen45 and the Xen63 it would follow logic that the larger internal diameter of the Xen63 would create less resistance to outflow compared with the Xen45 potentially resulting in lower post-operative IOP but also carry an increased potential for hypotony related complications for the same reason. Either way, the present study wasn't designed to compare the two devices nor patient selection for the devices.
the authors comment on the prior reports of the Xen63 being placed without the application of MMC but do not comment on the total dose used in patients in this series or how dose was determined.
Author Response
Reviewer #1
The current report presents the retrospective IOP and medication reduction data in 23 patients 18 months after undergoing an ab-interno placement of the Xen63 implant with subconjunctival MMC by two surgeons. The report demonstrates a sustained IOP lowering and reduction in IOP lowering medications that was sustained at 18 months post-op with a rate of secondary glaucoma interventions similar to other Xen device studies.
Thank you very much indeed for this comment.
The paper has several punctuation and grammatical errors related to translation that require profession English editor review and editing.
The English has been carefully reviewed
- Materials and Methods:
Patients: Wording of the inclusion and exclusion criteria contains a long run-on sentence. This makes the criteria used by the authors for patient selection confusing and unclear.
It was changed to: “Patients included in the study were aged ≥ 40 years; had a clinical diagnosis of glaucoma, and an unmet target IOP despite medical therapy. Patients with narrow-angle glaucoma were included if, in surgeon opinion, there was enough space to im-plant the device safely. Patients with closed-angle glaucoma; severe conjunctival problems; phacodonesis; progressive retinal or optic nerve disease of any cause; or history of major ocular surgery (except phacoemulsification) within the previous 6 months were excluded from the study”.
- Surgical Technique
The authors do not present the dose of MMC administered, only the concentration of MMC solution injected into the subconjunctival space. MMC dose would be important in comparison of other published data on the Xen device.
The following sentence was added: “(0.1 mL of mitomycin C 0.02%-0.03%)”
- Results
- 1Figure 3 Kaplan-Meier analysis should have a title and description added similar to the other figures and table.
The following information was added: “Figure 3. Kaplan–Meier survival curve for complete success. Complete success occurred in 77.8% of eyes, while qualified success occurred in 94.4%”.
3.2 Safety
Total number of patients requiring a secondary intervention should be reported. According to table 4, secondary interventions may have been required in 14 of the 23 patients. This would represent 61% of patients requiring a secondary intervention after initial Xen63 placement. The authors should further discuss the high rate of needing additional interventions or if a few patients required several of the secondary interventions reported.
We do not agree with the reviewer comment. Additional intervention in any glaucoma filtering procedure is a new surgical procedure and needling is not considered as an additional procedure but rather to be “part” of the normal bleb management (similar to YAG laser suturelysis for trabeculectomy). Similarly needling is not considered a failure in any of the previous XEN papers and using this anomalous criterion would make impossible to compare our paper with previous ones.
- Discussion
First paragraph briefly discusses the management of IOP lowering with drops followed but surgery but neglects to mention SLT as an additional option in management.
Laser therapy was added.
4.1 On line 246-248, the authors state "This may be an important point, as the larger caliber of the Xen 63 device could by clinically meaningful for patients with high preoperative IOP." This statement should be removed as the present study was not designed to compare results with the Xen45. If there is a difference in the results between the Xen45 and the Xen63 it would follow logic that the larger internal diameter of the Xen63 would create less resistance to outflow compared with the Xen45 potentially resulting in lower post-operative IOP but also carry an increased potential for hypotony related complications for the same reason. Either way, the present study wasn't designed to compare the two devices nor patient selection for the devices.
It was done.
4.2 The authors comment on the prior reports of the Xen63 being placed without the application of MMC but do not comment on the total dose used in patients in this series or how dose was determined.
In the methods section it has been mentioned that: “Mitomycin-C (MMC) (0.1 mL of mitomycin C 0.02%-0.03%) was injected intra-tenon in the supero-nasal quadrant prior to surgery”.
Mitomycin-C dose was
The dose of mitomycin was selected according to the surgeon criteria.
Reviewer 2 Report
The authors describe a retrospective study that evaluated the effectiveness of Xen65 implant in glaucoma patients at 18 months post-surgery. The use of scatterplots, survival curves and a combination of upper IOP limit and % reduction for defining success are commendable. I had the following concerns/questions:
1. Were the 4 eyes that needed additional surgeries considered as failures? If not, they should be. They should also be included in the denominator for determining the success percentages (if not done already)
2. The "Safety" section is repeated.
3. Any reason why the number of classes of meds decreased between 12 and 18 months?
Author Response
Reviewer #2
Comments and Suggestions for Authors
The authors describe a retrospective study that evaluated the effectiveness of Xen65 implant in glaucoma patients at 18 months post-surgery. The use of scatterplots, survival curves and a combination of upper IOP limit and % reduction for defining success are commendable.
Thank you very much indeed for this comment, we highly appreciate it.
I had the following concerns/questions:
- Were the 4 eyes that needed additional surgeries considered as failures? If not, they should be. They should also be included in the denominator for determining the success percentages (if not done already).
It was done. As a consequence, table 2 has been modified and figure 3 (Kaplan-Meier curve) has been adapted.
- The "Safety" section is repeated.
It has been corrected
- Any reason why the number of classes of meds decreased between 12 and 18 months?
After reviewing the data, we have not found any apparent reason. As an explanation, this decrease in the number of ocular hypotensive medications might be a consequence of trying to withdraw treatment in patients with good IOP control.
Round 2
Reviewer 2 Report
Review questions were addressed well.